# Audio Storytelling Innovation in a Digital Age: The Case of Daily News Podcasts in Spain

**Leoz-Aizpuru Asier** [1] and **Pedrero-Esteban Luis Miguel** [2,*]

1    Faculty of Social and Human Sciences, University of Deusto, Camino de Mundaiz 50, 20012 San Sebastián, Spain; asier.leoz@deusto.es
2    Faculty of Communication and Arts, University Antonio de Nebrija, Hoyo de Manzanares, 28040 Madrid, Spain
*    Correspondence: lpedrero@nebrija.es

**Abstract:** On the 1st of February 2017, *The New York Times* published the first episode of 'The Daily', a news podcast hosted by Michael Barbaro that, five years later, has become the most popular in the world with four million listeners each day and more than 3000 million accumulated downloads. The unprecedented success of this audio format, that has emerged in a traditional newspaper, has helped to revamp radio news and has spread in various versions all over the world. This investigation analyses daily podcasts in Spain and, by means of a quantitative and qualitative study, identifies their main themes, narrative structures, and expressive contributions based on four illustrative experiences in this market: 'Quién dice qué', 'AM', 'El Mundo al día', and 'Un tema al día'. The results reveal the consolidation of two clearly defined models: a more conventional one based on radio bulletins and news reports; and another, more innovative model that replicates the anglophone formula that opts for depth, dissemination, and a conversational tone to redefine the canons of the audio news narrative and take advantage of the potential of audio as a new distribution channel for newspapers in the digital eco-system.

**Keywords:** podcast; daily news; audio news; storytelling; innovation; information; format

## 1. Introduction

From the time that they first appeared, newspapers have often been named depending on how often they were published or distributed: this was the case with the *Daily Cur-ant* (1702) and *The Weekly Review* (1704) in England or *The Boston Newsletter* (1704) in the USA [1]. In this respect, the term 'daily' merely refers to anything that appears each day regardless of its content and format. However, the emergence of the daily news podcast and the global success of 'The Daily' (*The New York Times*, USA) or 'Today in Focus' (*The Guardian*, UK) have given this name a specific meaning.

According to Newman and Gallo [2], a 'daily' provides online audio news content broadcast at regular intervals (usually Monday–Friday), and has a short duration (3'–30') and a personal narrative treatment linked to its host, who tries to emulate the identity of the medium that produces it. The similarity of the format to traditional radio is inevitable: on the hundred-year-old radio medium there have always been well-told stories and reporting or chronicles able to transport the listener to other places or states of mind, but not in the news genre. This is what makes it different: it lies in the combination of two formats: news (interview, column, and piece) and narrative (reporting or documentary with direct participation by the storyteller), aimed at informing people, that is, at explaining what is happening, which is the fundamental basis of journalism [3]. In actual fact, the daily is not innovative due to its content, but due to its approach: its main contribution is that it has reinterpreted how news is narrated on the radio.

The news podcast has emerged to explain the key elements in the news of the day in view of the saturation of news and distribution channels [4], and of the speed that

the Internet and the social networks have provided for journalism that needs to reconcile itself with the essence of the profession. This is why the daily helps to recover the credit that the news media have lost [5] and also to meet a need that linear radio cannot always address: offering clarity in chaos. Faced with the frenetic pace of news coverage, podcasts provide focus and context, which contrasts with the standardisation of programming and the regression in the creative level of content that radio has undergone in periods of great technological change [6]. As the specialists stress, the podcast transforms information into knowledge and adds the subjective experience of its author to this: the live show answers the question, "what's happening"; radio on-demand is a response to "what have I missed"; the podcast, to "what can I learn" [7].

Martínez-Costa and Lus-Gárate mark the beginning of daily news podcasts in 2006, with the launch of 'Newsdesk' by the British newspaper *The Guardian* [8]. Since then, as Newmann and Gallo conclude in a report published in 2019 by Reuters Institute—based on the analysis of 59 news podcasts from the United States, the United Kingdom, Australia, France and Sweden—three main variants can be categorised [9]:

(a)     Micro-bulletins: short news bulletins lasting just a few minutes that aim to provide a quick summary of the day's news. Examples: 'BBC Minute' or 'NPR News Now'.
(b)     News round-ups: these are longer podcasts that have the aim of briefing people at particular points in the day with a short update. Example: 'FT News Briefing'.
(c)     Deep-dive analysis: these typically analyse one story in greater depth. Example: 'The Daily'.

'The Daily' is precisely the news podcast that has now become a worldwide benchmark. It was first broadcast by *The New York Times* (NYT) on the 1st of February 2017 with the following description: "This is how the news should sound. Twenty minutes a day, five days a week, hosted by Michael Barbaro and powered by *The New York Times* journalism". As well as revindicating the profession, emphasis is also placed on the figure of the host, who invites listeners to a restricted space—the newsroom—to tell them about current issues. The term 'host' entails warmth and proximity, becoming extremely important due to his/her close relationship with an audience that gradually becomes a community.

This explains how the NYT's pioneering daily, through its empathetic host, Michael Barbaro, offers not only facts, but also feelings [10]. 'The Daily' emerges from the dialogue that the journalist has with his colleagues at the newsdesk, who he asks to explain a specific issue that they know really well as they have written about it in the newspaper. In less than two years, the NYT podcast had more subscribers than the newspaper itself, had a billion downloads [11], and generated 400,000 dollars a month for the emblematic head. It also helped to rejuvenate the target audience of subscribers, many of whom do not consume any other product of the brand. All this is due to a team of 18 people (in 2021 they increased to 39) with a background in audio who provide the space with a unique sound identity, ranging from the theme song specially composed for this podcast to the locution and other details of the narrative construction [12].

Each morning at 6 a.m. 'The Daily' appears on Spotify and the *The New York Times* web page, where it features prominently. It offers a conversational programme lasting about 20 min in which Barbaro interviews someone from the newsroom who is a specialist in the subject (sometimes there are two) and which is rounded off by a final one-minute long section with some short news items. "Here's what else you need to know today" is the way that they are presented. NYT journalists with specific knowledge about each subject take part in each episode.

The success of *The Daily* in 2017 led to various replicas in newspapers all over the world, especially in the US and the UK (Figure 1). *The Guardian*, the British daily that coined the term 'podcasting' in 2004 [13] and had launched a podcast from 2006 to 2010 with the main contents of its print edition—*Newsdesk*, later renamed *Guardian Daily*—relaunched it in November 2018 with the name 'Today in Focus'. In December, *The Washington Post* launched 'Post Reports', a daily evening edition (17:00), and the *Financial Times* did the same with 'News Briefing'. Other prestigious publications joined a trend that other news

media have also adopted: radio (NPR, iHeartRadio, and BBC), television (CNN, ABC, and NBC), and the Internet (Vox and Axios).

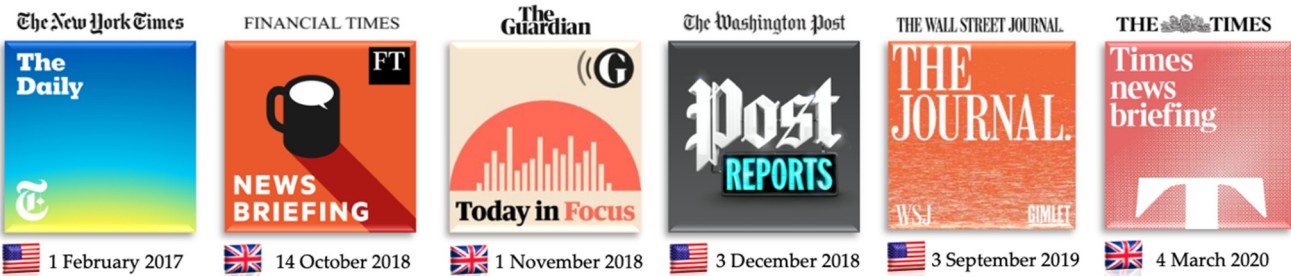

**Figure 1.** Daily podcasts produced currently by the most important newspapers in the USA and UK. Source: Prepared by the author.

In Spain, the early experiences with news podcasts go back to May 2018, when the *El País* daily, the leading newspaper among the Spanish press and one of the most prestigious both in Spain and abroad, began to produce 'El País Noticias'. This was a three-minute bulletin with the most important issues of the day. Six months later, in November 2018, *Las Noticias de ABC* started up. This was another micro-bulletin lasting two to three minutes with a summary of the latest news in Spain and the rest of the world through headlines from the political, economic, and international sphere, as well as sport, social life, and digital and cultural affairs. According to its producers, this podcast aimed to provide information, opinions, and trivia, as well as the most important features and exclusives from *ABC* [14]. As a technological innovation, you could listen to it not just on the web or the main audio platforms, but also through the brand new Amazon (Alexa) and Google (Google Home) smart speakers. Furthermore, in contrast to the morning edition of *El País*, *ABC* offered two editions, one in the morning and one at night (except on Fridays, Saturdays, and Sundays).

Unlike the anglophone dailies, the news bulletins of *El País* and *ABC* are also broadcast at the weekend with subjects from the supplements of both newspapers. The same thing happens with the news summary by the *La Razón* daily that, since the summer of 2021, has published the news podcast, 'Buenos días': this runs for five minutes and provides the keys to starting the day well informed. This variant matches the second of the three types categorised by a report from the Reuters Institute, a summary (b), longer than the micro-bulletin (a), but less crafted than the deep-dive analysis (c) that 'The Daily' has opted for since it began. This model was the one that was launched in June and September 2021 by two highly important newspapers on the market: *El Mundo* (a daily with print and digital versions) and *elDiario.es* (an online native medium). They were joined in November by another exclusively digital daily, *El Debate*, whose podcast is broadcast from Monday to Friday at night to provide an assessment of latest news [15].

In the middle of this timeline of events, when none of the Spanish news media had yet opted to produce a daily, three audio streaming platforms—Audible, Spotify, and Podimo—at almost the same time (October–November 2020) exclusively launched another three titles that introduced this format onto the market: 'Quién dice qué', 'AM', and 'La vuelta al día' (Figure 2). These began to compete with radio for information on current events through the use of new narratives and expressive approaches that are analysed in this study according to the methodology described below.

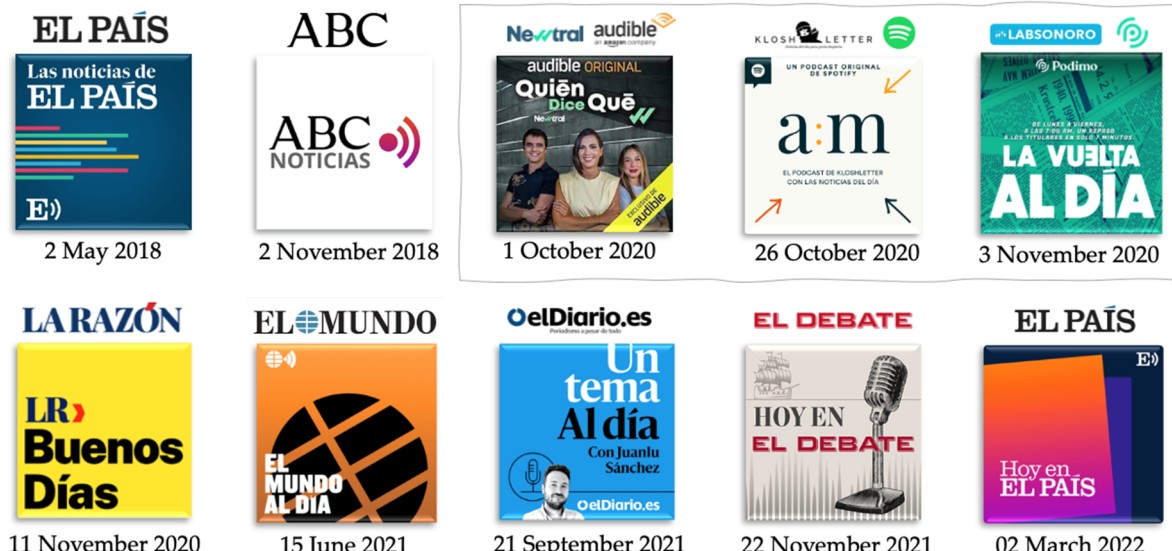

**Figure 2.** News podcasts of Spanish newspapers and audio platforms. Source: Prepared by the authors.

## 2. Materials and Methods

The aim of this research is to answer the following questions: (1) What are the models of daily podcasts in Spain (micro-bulletin, news summary, or deep-dive analysis)? (2) What are the differential aspects in the language and form of expression used to deal with the news content? (3) Do news podcasts in Spain renew the narrative and sound design of radio news programmes? In order to assess the differential aspects in the audio narrative offered by daily podcasts in Spain, we have analysed four representative podcasts in depth: on the one hand, the variants set up by platforms (pioneers in their commitment to this format) and, on the other, those produced by both traditional and digital news media: 'Quién dice qué' (Audible), 'AM' (Spotify), 'El Mundo al día' (*El Mundo*), and 'Un tema al día' (*el diario.es*).

### 2.1. 'Quién Dice Qué', the First Daily News Podcast

The 'Quién dice qué' daily was started and run by the journalist Ana Pastor who, in January 2018, founded Newtral, an audiovisual content startup that works in three areas: the production of television programmes and new narratives on social networks; innovation in journalism through fact-checking; and research using Artifical Intelligence protocols. The idea was to produce an Anglophone-style daily, with only a few news items but covered in some depth. To do this, she signed up two young journalists, Jacobo Pedraza and Marina Vázquez, with experience in musical radio and TV, who were asked to prepare the news using accessible language. The Audible platform presented it like this: "Current affairs are something more than just the very latest news: they are the premise that 'Quién dice qué' is based on, a current affairs programme for millennials, produced by millennials. Quite different from traditional news programmes".

During its first month, the podcast was produced in a radio studio and was the same as the traditional radio format: after an opening ident, the hosts addressed five news items in an informal tone, for which they interviewed journalists from the Newtral team itself near-live. Each episode lasted about 23 min. However, after the first 30 editions, the daily took a break for a month and came back with a new look: shorter—12 min—and with a monographic approach; in addition to this, the programme was now produced in the hosts' home in Madrid. It opted to get closer to listeners by employing a more leisurely spoken pace and a more intimate tone, as well as a more carefully prepared audio design: the sound levels acquired narrative importance and were embellished with archive recordings, which gave the podcast a new identity.

Each episode contains extracts from one or two interviews with experts with meticulous editing. 'Quién dice qué' is formally presented with a title that includes a concept and a question. For example: "15-M: What was it and what's left of it?" (15 May 2021); "Smoking: how many people smoke and how many manage to give it up?" (29 May 2021), "Female day labourers: What's going on in Huelva?" (5 June 2021), and "Sitting for public entrance exams: is this the only way for young people to find a decent steady job?" (27 October 2021). The programme is produced on the day before it is streamed so that the podcast can be available on the Audible platform each morning at 7 a.m. This service is only accessible by making a monthly payment of 9.99 EUR.

## 2.2. 'AM', the Audio Adaptation of the First Newsletter in Spain

'AM' is the podcast of *Kloshletter*, a newsletter founded in 2017 by the journalist, Charo Marcos. With extensive experience on the web page of the *El Mundo* newspaper and on RTVE, Marcos wanted *Kloshletter* to have an audio format to complement the written version. Inspired by e-newsletters in the USA, behind this daily summary there is a vindication of the work of crafting a newsletter, a third-party proposal through journalistic curation that an algorithm cannot provide. "My journalistic training and experience in the media helps me to curate information and tell things in a neutral pleasant way so that whoever reads this summary can develop an informed opinion" [16].

The daily was launched on the 1st of October 2020 with two hosts, Eixchélt González and Jon Elícegui. Both of them prepare each episode from their home in the city of Valencia, guided by Charo Marcos from Madrid. The script is written after a comprehensive reading of the digital press of the day, paying special attention to the local press. It is recorded at 6:30 in the morning and edited afterwards so it can be available on Spotify at 8:00 a.m., at the same time as *Kloshletter* is published (with the complete podcast transcription).

Since episode 6 ('Intrigues', 19 August 2020), each edition of 'AM' has had a title, the same as *Kloshletter*. This is a short phrase based on a play on words that refers to the news item of the day using the title of a film or a set phrase. For example, 'Ella baila sola' (She dances on her own) (5 May 2021), the name of a musical duo, refers to the news of Isabel Díaz Ayuso's great victory in the autonomous elections in Madrid. Other examples are 'The unswept house', 'Here, to stay', 'The table wobbles', and 'The house of trouble'. The headline always refers to the main news story of the day, which is tackled at the beginning, immediately after another shorter news item.

On average 'AM' lasts for 6 min. It offers five news items—each one lasts for a minute—and it does not cite any sources, although they do appear as links in *Kloshletter*. The hosts' voices alternate headlines and contents with hardly any interaction between them. There are no other voices in the piece, and voice clips are only used occasionally. The theme tune is instrumental with a lively rhythm and alternates with musical padding. The language is informal, with colloquial turns of phrase and direct appeals to listeners, with regular calls to their Twitter account to complete the information. The tone is warm and intimate towards listeners, who the hosts address informally. Visually, the podcast cover shows a still treated with a different colour each day of the week.

## 2.3. 'El Mundo al Día', the First Daily Podcast of a Newspaper in Spain

'El Mundo al día' is the first deep-dive analysis daily news podcast created by a press publication in Spain. It was first broadcast on the 15th of June 2021 with the episode entitled 'Europe takes its mask off (and who Sergio Ramos is)', although two weeks before this it had launched the episode, 'A day is a day' in its trial phase. Its host is Javier Attard, a journalist with prior radio experience, and is produced in the *El Mundo* newsroom (a detail that the host reminds us of in each episode). Each edition lasts about 16 min, and it can be listened to on the newspaper's web site at 7.00 a.m., as well as on Spotify, Apple Podcast, and iVoox.

One or two people from the *El Mundo* newsroom regularly take part in each edition. The subjects they choose match the ones that form part of the print version, sometimes as

a main news item and on other occasions as an exclusive: Selectivity exams, the plans by the Government to move ETA prisoners closer to the Basque Country, increased electricity charges, temporary contracts, the demonstrations against the pardons for Catalan prisoners, the elections in Madrid, the law on euthanasia, the fall of Afghanistan, or the working conditions of delivery workers. The podcast has an illustrative title ('Germany under water', 'Things we do not know about the fifth wave', or 'Ciao, Raffaella: the most Spanish Italian'), but it commonly resorts to humour or irony in a product that is very much in line with the newspaper it is linked to, which is usually critical of the government, running headlines such as: 'Cessions by Sánchez S.L: MIR destination Catalonia', 'the tale of the president who wanted to be king (for holidays)', 'Spanish Sanchist Workers' Party', or 'This is what the housing law is like: what can go wrong?'.

The structure of this podcast varies depending on the news that it chooses: in most cases, there is a single subject that takes up all the time; in others, the main news story does not require such extensive development and two or even three subjects are addressed, always by the journalists from *El Mundo* that publish the texts that provide the basis of the daily in the newspaper or in its supplements. These contributions are recorded by phone, near live, or by voice mail. Since it appeared, 'El Mundo al día' has been the first and closest adaptation in Spain of the Anglophone format of the daily. In February 2022, eight months after it first came out, had more than a million reproductions of its episodes.

Its visual identity is striking, as in its initial phase (15 June to 19 September 2021) it was identified by a logo and intense corporate tone (dark orange); however, since the 20th of August 2021, the logo has been superimposed on an image in colour that aims to sum up the issue in question; this additional change would be adjusted a few days later, with the episode 'The hole in your pocket made by the most socially engaged Government' (31 August 2021); from then on, 'El Mundo al día' redefined its graphic appearance, which is now based on an image that refers to the news story in question treated with a filter with the same tone and the name of the podcast superimposed in white (Figure 3).

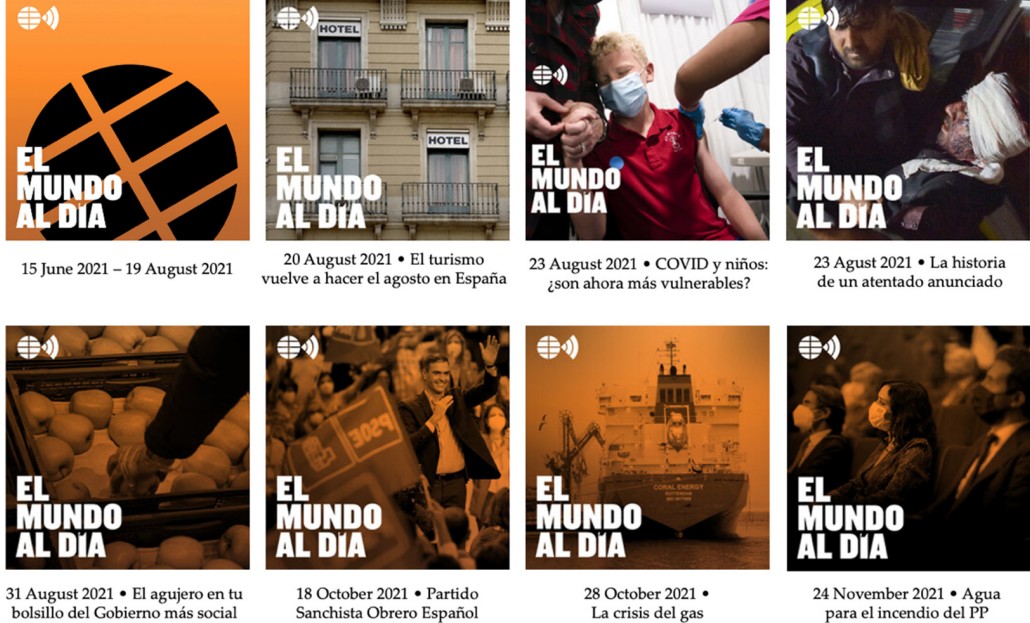

**Figure 3.** Evolution of the visual aesthetic of the covers of 'El Mundo al día'. Source: *El Mundo*.

### 2.4. 'Un Tema al Día', the First Daily Podcast of a Native Digital Newspaper in Spain

On the 20th of September 2021 the sub-director of *elDiario.es*, Juanlu Sánchez, announced on the digital newspaper's web site the arrival of a daily news podcast. It was called 'Un tema al día', and would complement the newsletter, 'Al día', that he was already

the editor of. His aim was to "explain current affairs . . . an audio pill to accompany you at breakfast, on the way to work, in your lunch break or before you go to sleep" [17].

Each morning since the 21st of September, 'Un tema al día' has reached the digital platforms at 6:30 as a complement to the newsletter. The podcast is sponsored by Podimo and subscribing to this platform or to *elDiario.es* enables you to listen to it the night before. To address the subject chosen in each episode, the host—Juanlu Sánchez himself— interviews one or two of his colleagues from the newsroom as well as, on occasions, experts who do not form part of the staff.

The first subject that this daily dealt with was the eruption of a volcano on the island of La Palma which grabbed the news spotlight in Spain for several weeks. Since then, the subjects addressed on 'Un tema al día' have dealt with a wide variety of subjects and sections including, among others, the sudden outage experienced by WhatsApp, Facebook, and Instagram in October, the climate summit in Glasgow, the new Rent Act, the metalworkers strike in Cadiz, and the rise of the far right in Spain. Just like on 'El Mundo al día', these are always issues dealt with in the newspaper edition, and can count on the journalist responsible for the original piece. In the same way, the editorial line of *elDiario.es* can be seen in the subjects chosen and how they are dealt with.

Each episode of 'Un tema al día' lasts for between 8 and 12 min and deals, as its name indicates, with a single issue. It is based on the conversation recorded with journalists from the editorial office and specialists, and is sonically embellished with background music recorded at the time that the interviews took place, and, on occasions, with film clips or archive recordings requested from radio stations as well. As for its visual identity, this is provided by a still, with a blue background, of the host of the podcast accompanied by letters superimposed in black and white.

*2.5. Methodology*

In order to answer research questions previously mentioned, we have turned to an analysis methodology that combines the quantitative and qualitative methods of the four programmes that were chosen: P1 ('Quién dice qué'), P2 ('AM'), P3 ('El Mundo al día'), and P4 ('Un tema al día'). On the one hand, programmes broadcast in different months—May, June, September, and October 2021—were selected and listened to closely in order to avoid any possible bias linked to the latest news. In order to apply the analysis to a homogenous sample, the same number of episodes (28) of the four news podcasts were chosen at random as shown in Table 1.

**Table 1.** No. of episodes analysed of each news podcast selected as a sample.

|  | **P1: QDQ** | **P2: AM** | **P3: EMD** | **P4: UTD** |
|---|---|---|---|---|
| May | 7 | 7 | — | — |
| June | 7 | 7 | 10 | — |
| September | 7 | 7 | 10 | 12 |
| October | 7 | 7 | 8 | 16 |
| Total | 28 | 28 | 28 | 28 |

By using Trint voice recognition software, the audios have also been transcribed in order to identify their narrative construction and to assess the clarity and expressiveness in the way that they are drafted. Finally, with regard to the analysis variables, the following characteristics of daily podcasts have been compared that were defined as specific and differential in previous academic studies [18]: (a) the figure of the host; (b) the contents; (c) the narrative resources; (d) the identity and brand; and (e) advertising and promotional content (Table 2). An analysis and comparison of these aspects among the cases that have been studied are provided in the following sections.

**Table 2.** Formal variables defined to analyse news podcasts.

| Name | Premiere | Production | Length | Host | Distribution |
|---|---|---|---|---|---|
| 'Quién dice qué' | 1 October 2020 | Newtral | 12′ | M. Montalbán/J. Pedraza | Audible |
| 'AM' | 8 October 2020 | The Voce Village | 6′ | E. González/J. Elicegui | Spotify |
| 'El Mundo al día' | 1 June 2020 | Unidad Editorial | 15′ | Javier Attard | Multiplatform |
| 'Un tema al día' | 21 September 2020 | elDiario.es | 12′ | Juanlu Sánchez | Multiplatform |

## 3. Results

### 3.1. Figure of the Host

In the four daily podcasts the importance of the host can be seen. In three of them—'AM', 'El Mundo al día', and 'Un tema al día'—his/her role is extremely important; they introduce themselves at the beginning ("I am Eixchélt González, and I am Jon Elícegui", "I am Javier Attard"), when the usual case in radio bulletins is for this to be performed by a recorded call sign [19]; they are on familiar terms with listerners and address them with a feeling of closeness and a style of speaking that transmits warmth. If we take the anglophone model as a reference, 'El Mundo al día' and 'Un tema al día' are the podcasts that are closest to each other as far as the function of the host is concerned. Javier Attard and Juanlu Sánchez address listeners informally and speak to them in an intimate tone at a leisurely pace; this is accompanied by an explanation in a didactic style of the ins and outs of the subject in question: in the style of Michael Barbaro in 'The Daily', both stress that the programme is created in the editorial office of the newspaper and even share the sound of the actual newsroom when they drop by to interview colleagues from the newspaper. The dialogues with the latter also reflect the informal tone used with people dealt with on a daily basis.

This feeling of closeness is also a characteristic feature of the hosts of 'Quién dice qué' and 'AM', who employ a style of language aimed at those young listeners that they identify with and use expressions that close the gap between them. An example of this is the episode of 'Quién dice qué' about the World No-Tobacco day, in which the host (Jacobo Pedraza) confessed that he himself started smoking when he was 24, or the episode of 'AM' when Eixchélt González, as her colleague Jon Elícegui has lost his voice—he makes an attempt to say hello—starts by saying: "Jon Elícegui has lost his voice today, so it's just you and me". Although the interaction between the hosts is minimal, they display aspects of their daily life to the listener, reveal part of their identity and idiosyncrasies, and convey emotions and not just data [20].

### 3.2. Duration and Contents

Three of the four podcasts that are analysed ('El Mundo al día', 'Un tema al día', and 'Quién dice qué') opt to deal with a single story in depth for 12–15 min with the help of guest voices: journalists from their own newsroom. For its part, 'AM', more like a news summary, addresses five issues and lasts for 6 min on average. As for the subjects that are dealt with—that is, the sections of a daily in its audio version—we can see that these are mainly Spanish political issues to do with the pandemic, such as the anti-COVID measures or questions concerning vaccination. In the national politics section, the controversy over the pardons for the defendants jailed due to the Catalan independence referendum in 2017 and the effects of the autonomous elections in Madrid in May 2021 have taken up most of the time on the podcasts that have been studied. Furthermore, the eruption of a volcano on La Palma (Canary Islands)—and follow up and consequences—has been the most frequently discussed issue apart from the two aforementioned sections.

As shown in Figure 4, in the period on which the research focused—May to October 2021—the topics that dominated the daily news podcasts were those related to national politics, ahead of news related to the pandemic and the measures aimed at tackling it in Spain. The other three most frequently covered topics in the titles analysed were international politics, (US–Mexican relations, general elections in Germany, etc.), the economy

(new rent law, the emergence of bitcoin as an official currency, programmed obsolescence, etc.), and events, almost all of which were related to the eruption of the La Palma volcano in the Canary Islands.

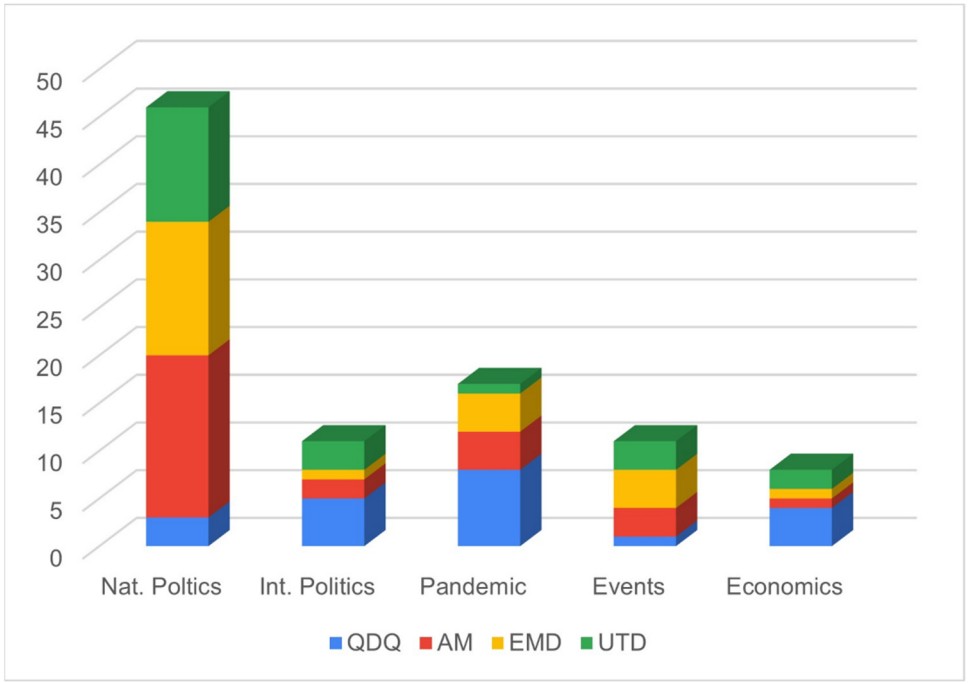

**Figure 4.** Sections of the four podcasts analyzed. Source: Prepared by the authors.

*3.3. Narrative Sound Resources*

The four podcasts studied here are backed up by a wide variety of sound resources that each of them creates and uses in order to achieve a differential narrative. 'El Mundo al día' and 'Un tema al día', in which the music, sound effects, film clips, and audio archives spice up and make the storytelling significantly more dramatic, stand out.

3.3.1. Theme Tunes and Background Music

The use of two theme tunes—an electronic one with a lively rhythm to open and another soothing theme to close—shows the interest that the 'El Mundo al día' daily has in making the soundtrack an additional narrative element, as well as inserting other melodies to build up tension in the story. The music acquires a clear editorial purpose through its extra-diegetic use, so that, the 'The soap opera of the pardons provokes the employers' episode (21 June 2021) starts with an opera piece that defines the press conference that Pedro Sánchez had given at the Liceu Theatre in Barcelona. Along the same lines, the 'Summer of lovelessness' episode (27 July 2021) on the differences between the conservative parties, the PP, and Vox, begins with Amercian music from the 1960s that evokes the height of the hippy movement in San Francisco in 1967.

Although music is a production aspect that makes it possible to standardise and identify contents, and to structure and link the storytelling on traditional news radio [21], the interplay of levels and the acoustic treatment of the pieces acquires a different dimension in the 'El Mundo al día' daily: this is shown by details such as the echo effects as sudden warnings, or the contrasts such as the one in the "Face masks: will we be able to say goodbye to them soon?" episode (16 June 2021), which begins with a poorly modulated voice that, just seconds later, sounds clear and bright, and simulates the relief of removing a facemask. In 'AM', pieces of music are also introduced which have a semantic load, but what is most striking is the use of effects such as bursts, camera clicks, and the psst effect when opening something, which reinforces the podcast slogan (the news of the day for people who are wide awake).

### 3.3.2. Background Noises and Effects

In 'El Mundo al día' and 'Un tema al día', the use of background noise that is typical in radio reporting is striking. For example, when talking about the volcano in eruption on the island of La Palma in the Canaries, the sound of the crater and the lava proved to be so shocking and mesmerising that even the TV stations offered several seconds of uninterrupted coverage. In the '39 days at the foot of a volcano' episode (28 October 2021) of 'Un tema al día', this sound remained in the background all the time and even closed the programme as wordless credits. For its part, 'El Mundo al día' in the 'Additionally, the lava reached the sea" (29 September 2021) episode offered the roar of the volcano captured from the Ramón Margalef boat of the Spanish Institute of Oceanography.

The edition of 'El Mundo al día' on the 15th of June turned to a revealing sound montage: it was devoted to the meeting between Pedro Sánchez and Joe Biden at the NATO Summit in Brussels, which the Spanish prime minister had announced would be long and yet hardly lasted for half a minute. The episode begins with the audio of the TV commentary on the 100 metres race in which Usain Bolt won the gold medal in London 2012. This resource served as a pretext for Javier Attard to humourously stress how short the meeting was, which also covered about 100 metres. In other cases, by introducing his editorial colleagues (especially if they are correspondents in other countries), the conversation is dramatized by the effect of a phone call. The degree of sophistication in the narrative style of the reporting can be seen in the 28 episodes that have been analysed.

### 3.3.3. Documentary Archives and Film Clips

Both daily podcasts make expressively effective use of film clips. 'El Mundo al día', with the title of 'A day is just a day' (01 June 2021), featured dialogues from *Dead Poets Society* (Peter Weir, 1989) in an episode devoted to education, whereas 'Un tema al día' opted for film dialogues in the 'You cannot tell anyone this' episode (27 September 2021), which illustrated the 'off the record' journalism with a clip from *All the President's Men* (Alan J. Pakula, 1974) where Robert Redford and Dustin Hoffman talk in a dark carpark.

The use of archive sound proves to be a remarkable device in 'El Mundo al día' and 'Un tema al día', and is also striking as these are new products that do not have a documentary sound archive of their own. The use of recordings—personal accounts and background noise—from Radio Lorca Cadena SER in the 'La Palma-Lorca, a journey in time' episode (23 September 2021) of 'Un tema al día' to compare the earthquake in Lorca in 2011 and the volcano on La Palma in 2021, or the use of recordings from the RAC1 station from the 1st of October 2017 in the 'Four years since the 1-O' episode (01 October 2021) really stand out. In 'El Mundo al día', the sound archive is repeatedly used, ranging from the audios of the Olympics in London 2012 to the sounds of the Games at the opening ceremonies in Tokyo 1964 and Rio 2014 ('The Pandemic Games: dissection of Tokyo 2020', 22 July 2021) or the accounts from Radio Televisión Canaria in the 'Additionally, the lava reached the sea' episode (29 September 2021).

### 3.3.4. Switching to Personal Accounts and Recordings

Along with music, background noise, and film dialogues, 'El Mundo al día' and 'Un tema al día' are also innovative in the way that they present current affairs. So, both Javier Attard and Juanlu Sánchez often provide a voice clip explaining who the person that is talking is and in what context they are doing so, letting this clip run in the foreground only at important moments. This technique is an innovation compared to radio news, in which the edited voice clip is provided in its entirety without interruptions. The feeling it creates is interesting: it transports listeners back to what happened in the past as if it were happening at this very moment, which really captures their attention. An example of this is 'The abortion witches' episode (28 September 2021) from 'Un tema al día': the host talks about the parliamentary recording in which an MP insults someone and is called to order. The presenter tells us what has happened by lowering their voice and explaining in a confidential tone who the people are that can be heard in the clip; this style is unusual

in news reports or bulletins and is more associated with features, transferred here to the sphere of news. The same technique is repeated in the 'Isabel Díaz Ayuso's political ladder' episode (22 September 2021), where the host explains the context with the sound of Díaz Ayuso's speech after her victory in the autonomous elections in May in Madrid.

### 3.3.5. Location and Setting

Placing the listener in the newspaper's editorial office is also a powerful narrative device. They do this in 'El Mundo al día' and 'Un tema al día', where the setting, the actual newsroom, is presented and described in words and with background noise. In 'El Mundo al día', Javier Attard reminds us in each episode that he is speaking to us from the headquarters of the *El Mundo* newspaper, which reinforces his aim of telling "the news on the inside, what there is behind what is going on, what the job of journalists is really like". In 'Un tema al día', sounds can be heard captured in the actual *elDiario.es* newsroom. An example of this is the 'Social democracy was not dead' episode (29 September 2021), in which Juanlu Sánchez walks, holding a recorder, to the desk where his colleague Icíar Gutiérrez explains to him what the core issues are in the elections in Germany. This way of introducing listeners to the person who is going to inform them is much more intimate than the traditional introduction on news radio, where we would hear something like "our colleague Icíar Gutiérrez is reporting on the elections in Germany. Hello, Icíar", following the model established by the traditional news bulletin [22].

### 3.4. Identity and Brand

This research confirms that news pocasts are linked to the newspaper brand that produces them so that they help to reinforce their prestige and to extend and diversify their scope. This feature can be clearly seen in both of the daily podcasts produced by press publications—*El Mundo al día* and *Un tema al día*—and even in the bulletins by *El País* and *ABC*, although in this case their narrative approach is not innovative. These audio contents create synergies with their respective newspapers and encourage listeners to read print or digital information, whose authors directly participate in their audio version.

'Quién dice qué' also projects the Newtral brand by inviting its journalists to 9 of the 28 episodes analysed, and by stressing the relationship of the podcast with the producer during the identification announcements (twice in each edition). 'AM' is also linked to a newspaper brand: as well as being mentioned during the identification announcements (two per episode), *Kloshletter* is published at the same time as the podcast, includes its literal transcription and creates a feeling of closeness to the host through an e-mail to subscribers that Charo Marcos closes with a "I will write to you tomorrow".

### 3.5. Advertising and Promotional Content

The advertising and promotional contents appear in a different way in each of the analysed podcasts. 'El Mundo al día' is produced by sponsorship: during the period studied, the British telecommunications company Vodafone and the Spanish company Telefónica were the commercial brands linked to this daily podcast, whose host, Javier Attard, mentions them at the end of the programme. In 'Un tema al día', Juanlu Sánchez makes an initial reference to the Podimo platform and it is mentioned again, in a woman's voice, before closing the episode. As for the podcasts streamed exclusively via the platforms that host them, 'Quién dice qué' and 'AM' expressly mention them (Audible and Spotify) at the beginning, and also do so with idents during the programme.

## 4. Discussion

The emergence of the Internet and the normalisation of mobile devices to access it have led to the convergence of languages and formats, the proliferation of means and points of access to contents, and the consolidation of new logics linked to digital transformation [23]. As a result of these changes, the information, entertainment, and relationship habits between users and the media—including the radio—have been redefined as they

have been forced to adapt their production, distribution, and monetization models [24]. The consolidation of the any-content, anywhere, anytime, any-device paradigm has the communicative contents on offer to an audience that freely enjoys the options of empowerment, participation and interaction provided by the digital world and the social networks [25].

It is in this context that the podcast has emerged, closely associated with the radio in view of the audio language that both communicative models share; however, the nature of podcasting has managed to renew the forms of expression on the radio and the way that it produces, distributes, and consumes information, entertainment, and fiction contents, especially among a younger audience. A quarter of the population of the US (80 million people) now listened to audio contents in this format on weekly basis in spring 2021; most of them are adults between 40 and 54 years of age (the so-called Generation X), but almost a third (31%) are Millennials (25–39 years of age), and a significant increase can be seen in the time that people over 55 years of age devote to podcasts (16% of the market). The most popular types of prgrammes are comedy (43%) and news podcasts (38%), followed by culture, true crime, education, music, and business [26].

Despite the greater penetration of the market by entertainment podcasts and talk shows, the use of this format to find out about the news or to learn about the key issues of the day can be seen more and more among young people: according to a survey by Pew Research in the US [27], one in every three adults from 18 to 29 years of age consumes news podcasts now and then, compared to 29% of the population between 30 and 49 years of age, 18% of people from 50 to 64 years of age, or 12% of people over 65 years of age. Listening to spoken audio content (radio, podcast, and audiobooks) has increased in the US by 40% in the last seven years (8% a year between 2014 and 2021), and the amount of time devoted to spoken contents now amounts to 28% of audio consumption; in fact, the time spent listening to podcasts has almost tripled in this cycle (8 to 22%), while the figure for the radio, which previously amounted to over two thirds of the total (79%), is now less than half (48%).

In Spain, people under 35 years of age were those who listened to podcasts the most (51%) in summer 2021 [28]. Compared to the rest of Western Europe, it is one of the countries where more internauts consume this format: almost 4 of every 10 (38%), above Norway and Sweden (37%), Italy (31%), France (28%), Germany (25%), or the UK (22%). The most popular subjects for Spanish users are science and technology, economics, and business or health podcasts, consumed by 15%, followed by current affairs (news, politics, and international events) with 12%.

So, audio on-demand is becoming a promising means to enable the traditional media, especially newspapers, to take advantage of the creative and expressive possibilities of this format to connect with the audience interested in current affairs in other languages and formats: daily podcasts provide new opportunities to tell the news by using different narratives, these may include useful contents that complement what the medium already offers in its written editions [29]. This strategy not only helps to retain their traditional audience, but also to attract new users who, in this way, can discover and access this brand. In short, it is a tool with huge potential in the commitment to digital transformation and consolidation of the news media [30].

## 5. Conclusions

Significant results have been obtained from the analysis carried out on the different types of news podcasts in Spain to assess not only how podcasting has progressed and matured, but also to evaluate its level of innovation with regard to what, up to now, has been one of the exclusive contents of radio: information on current affairs, which is still constrained by its production logics even in on-demand coverage [31]. Although differences can be seen when comparing these variants, the first conclusion that can be reached is that this environment is now competing with the hundred-year-old radio medium to provide sound contents with a certain degree of immediacy, especially when they are produced by journalistic media.

After studying the quantitative and qualitative variables defined in the research, the following answers can be given to the questions framed as the basis for this study. First of all, examples can be identified of the three models categorised as daily podcasts: (a) micro-bulletins ('Las noticias de El País', 'ABC noticias', and 'LR Buenos días'); (b) news summaries ('AM' and 'La vuelta al día), and (c) deep-dive analysis ('Quién dice qué', 'El Mundo al día', 'Un tema al día', and 'Hoy en El Debate'). The structure and aesthetic treatment turns out to be very similar in the variants of the first two models, whereas the third—which corresponds to the 'The Daily'—is where significant, innovative differences can be seen. Although 'AM' is also hosted by presenters who are identified by name and try to offer a less distant review of the latest news than in radio versions, neither the contents nor the narrative resources provide it with any differential expressive value. In fact, it lasts for no longer than 7 min, against the almost 15 min duration of 'El Mundo al día', 'Un tema al día', or 'Quién dice qué'.

The second question can be answered in a similar way: the identity of news podcasts compared to the radio in the selection and treatment of subjects is decisively confirmed in those produced by *El Mundo*, *elDiario.es*, and Newtral, which opt for editorial critera of their own, which quite often depart from the radio agenda when they refer to exclusive contents or alternative treatments of the media that produce the daily podcast. The host's personality, their own personal way of presenting information, and going deeper into the contexts of things by talking with their newsroom colleagues makes deep-dive news podcasts sound original, distinctive, and rejuvenating, thanks above all to the use of the elements employed in their expressive and aesthetic coverage. Their visual appearance is also notable, which in the case of 'El Mundo al día' is personalised by an image alluding to the subject dealt with in each edition.

In this respect, the third research question—do daily podcasts renew the narrative and sound design of radio news programmes?—complements and at the same time confirms the previous conclusion when we refer to the immersion model and its wide variety of sound resources. Extra-diegetic music, background noise and effects, film clips, superimposition of the host's voice over the testimony of interviewees, and dialogues among the editorial staff all give the storytelling an original narrative texture that is more direct and effective and less restricted by the traditional canons of expression in broadcasters' newsrooms, where live broadcasting factors encourage the use of genres with less complex production values such as news with or without recordings, chronicles, or interviews. [32].

These four podcasts form the first generation of a variant whose importance is becoming increasingly clearer on the digital audio market. This study not only reveals the potential that on-demand audio has as a vehicle to provide customised access to the latest news—a quality that radio on-demand also offers—but also shows the possibilities that the podcast offers newspapers to renew and open up the consumption of news contents to new audiences—especially to younger people—based on genres and narratives that are typical of the habits and demands of the digital ecosystem, including greater guarantees in their treatment [33]. As Michael Barbaro himself wrote by e-mail to the members of his team on the fifth anniversary of the premiere of 'The Daily', "the right combination of producers and editors, the right blend of audio journalists and storytellers, of composers and wordsmiths, Pro Tools wizards and guest whisperers—not to mention the world's best newsroom—could make a daily news podcast not just urgent and essential, not just beloved and addictive, but transcendent" [34].

**Author Contributions:** Conceptualization, P.-E.L.M. and L.-A.A.; methodology, P.-E.L.M.; formal analysis, P.-E.L.M. and L.-A.A.; investigation, P.-E.L.M. and L.-A.A.; resources, P.-E.L.M. and L.-A.A.; data curation, P.-E.L.M. and L.-A.A.; Writing—original draft preparation, P.-E.L.M. and L.-A.A.; writing—review and editing, P.-E.L.M. and L.-A.A.; supervision, P.-E.L.M. and L.-A.A.; project administration, P.-E.L.M. All authors have read and agreed to the published version of the manuscript.

**Funding:** This research received no external funding.

**Institutional Review Board Statement:** Not applicable.

**Informed Consent Statement:** Not applicable.

**Data Availability Statement:** The data are not available in a public place.

**Conflicts of Interest:** The authors declare no conflict of interest.

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
