# Peer review of "Audio Storytelling Innovation in a Digital Age: The Case of Daily News Podcasts in Spain"

_information, doi:10.3390/info13040204_

Round 1

Reviewer 1 Report

Overall, I appreciate the research topic, and there was clearly a great deal of work completed by the author(s). I would however, like to see what’s included in the conclusion section moved to the discussion. I understand scholars prefer various sequence formatting, but I believe the crux of the research relevance should be noted in the discussion section while the conclusion should be a very brief review of what was addressed. I would also like to see a future research section. This article could be the starting point of many more relevant podcast research efforts. 

Highlighting how a traditional mass medium is reaching younger audiences by reformatting itself into a digital audio format is a valuable topic of exploration. This is especially relevant given how so many younger individuals now turn to social media for news content. I see this article as a valuable contribution to podcast audience research

Author Response

We sincerely appreciate the assessment of the text and the reflections and suggestions for improvement. We consider it very relevant to highlight the relevance of this format so that audio content can reach more young people, and we have specified this in the conclusions. We agree that future lines of research are open, and this is noted at the end of the text.

Reviewer 2 Report

This is a very interesting article that studies an issue (daily podcasts) that has hardly been investigated from a scientific point of view, not only in Spain, but worldwide. Both the podcast and the daily format are phenomena in full expansion.

I suggest doing a final review of the English language and some additional bibliographical references are suggested as well.

  • Espinosa de los Monteros, M. J. (21.12.2018). Diez claves para analizar los ‘daily podcasts’. El País. https://elpais.com/elpais/2018/12/19/dias_de_vino_y_podcasts/1545214268_158714.html
  • Martínez-Costa, M.P.; Sánchez-Serrano, C.; Pérez-Maíllo, A. & Lus-Gárate, E. (2021). La oferta de pódcast de la prensa regional en España: estudio de las marcas centenarias de Castilla y León y Navarra. adComunica, 21, 211-234.
  • Martins, R. & Vieira, J. (2021). Podcasts in Portuguese journalism. The Case P24. Media Journalism, 21(1), 99-122. 
  • Newman, N. & Gallo, N. (2020). Daily News Podcasts: Building New Habits in the Shadow of Coronavirus, Reuters Institute for the Study of Journalism https://reutersinstitute.politics.ox.ac.uk/daily-news-podcasts-building-new-habits-shadow-coronavirus.
  • Reis, A. I. (2018). Invisible Audio: an Analaysis of the podcasts of Portuguese Newspapers. Lusophone Journal of Cultural Studies, 5(1), 227-243.
  • Schneier, M. (21.01.2020). The Voice of a Generation: Michael Barbaro Made the New York Times podcast The Daily a Raging Success. Or Is It the Other Way Around? New York Magazine. https://nymag.com/intelligencer/2020/01/michael-barbaro-the-dailypodcast-new-york-times.html

Author Response

We sincerely appreciate the evaluation of the text and the reflections and suggestions for improvement. We have sent the article for a second revision to refine the English translation, and we have also incorporated three of the six references you suggested, which greatly reinforce the bibliography that serves as a basis for this research.

Reviewer 3 Report

Introduction

Since the findings are similar to the study by Newman and Gallo (2019) and Martínez-Costa and Luz (2019), it is important that these studies are well referenced in the text. The conceptualization, characterization and typology of daily news podcasts offered by these articles can be further developed in the introduction.

Define what and which are the narrative formats as opposed to the news formats mentioned in lines 38 and 39.

The use of the term “radio à la carte” is not understood, as there is an English version already established. Delete this expression in line 53 and others.

Presentation of the material

Review the statement 'El Mundo al día' is the first daily news podcast created by a press publication in Spain (line 204), because there are antecedents in El País, ABC and La Vanguardia.

Review the statement 'Un tema al día', the first daily podcast of a digital newspaper in Spain. If you mean it is the first digital-native media, is correct.

Results

Complete figure 4 with the corresponding data so that it is easier to read, indicating the universe studied.

Discussion

Presents data on the popularization of podcast production and consumption. It is not directly related to the results offered. I suggest that this material be included in the introduction and previous context, and that the discussion be reworked by reviewing the results obtained in that initial context. And so, get the conclusions.

Conclusions

The conclusions are very descriptive, with the material obtained it is possible to advance in the analysis of the subject and open new approaches related to the editorial and visual identity of the daily news podcast with respect to its digital brand, the role of the distribution platforms in the popularization of the format, the presence of new actors in the production of journalistic content beyond digital media, etc.

On the other hand, with the available material it is not possible to answer question 2) What are the differential aspects in how information is processed? There are no in-depth interviews that help answer this question (how information is processed). I suggest  the reformulation of the question referring to the content analysis carried out.In addition, when authors mention in line 530 and 531 that "the identity of news podcasts compared to the radio in the selection and treatment of subjects is decisively confirmed in those produced", they must first explain the characteristics and identity of the radio news narrative, also present in question 3, but is not mentioned in the text.

Author Response

We sincerely appreciate the evaluation of the text and the reflections and suggestions for improvement. We have proceeded to implement them as follows:

  1. a) Introduction

The works of Newman and Gallo (2019) and Martínez-Costa and Luz-Gárate (2019) have been better referenced in the first part of the article. In addition, we have clarified the informative versus narrative formats (lines 38 and 39), described the main variants of the informative podcaste (lines 56-66) and replaced the term "radio à la carte" with "radio on demand", which is the consolidated term in this field.

  1. b) Presentation of the material

The reference to "El Mundo al día" as the first daily news podcast created by a press publication has been corrected: it is now indicated as the first deep-dive analysis podcast in Spain (the variant described above according to Newman and Gallo's classification).

The wording of section 2.4. has also been clarified to make it clear that "Un tema al día" is the first daily podcast by a digital native newspaper in Spain.

  1. c) Results

A paragraph has been added before figure 4 to explain the data collected in this graph.

  1. d) Discussion

It is considered that this section provides an interpretative context on the relevance of daily podcasts that makes more sense in this section, as the introduction and the previous context focus on the birth and development of the daily podcast variant.

  1. e) Conclusions

The formulation of question 2, which refers to the formal expressive treatment of daily podcasts, has been reworked, not to the processes of elaboration, the knowledge of which, as the reviewer rightly points out, requires other research techniques such as in-depth interviews.

The limitations of radio news narrative have been clarified to validate the argument that news podcasts renew the way of telling the news in relation to the Hertzian medium.